# Measurement Report: Effects on viability, culturability, and cells fragmentation of two bioaerosol generators during aerosolization of *E. coli* bacteria

Federico Mazzei[1,2], Marco Brunoldi[1,2], Elena Gatta[1], Franco Parodi[2], Paolo Prati[1,2], Virginia Vernocchi[2], Dario Massabò[1,2]

[1]Department of Physics, University of Genoa, Genoa, 16146, Italy
[2]Istituto Nazionale di Fisica Nucleare, Genoa, 16146, Italy

*Correspondence to*: Elena Gatta (elena.gatta@unige.it)

**Abstract.** Bioaerosol is a significant element of Particulate Matter (PM) and comprises various components, with bacterial species ranking as some of the most important. Reliable and consistent bioaerosol generators are essential for the investigation of bioaerosol in laboratory environments. Aerosol generators are utilized to evaluate the performance of bioaerosol collectors, explore the transport and deposition of biological particles, and study the health impacts and exposure to airborne microorganisms. The main goal of the bacteria experiments is to have an aerosol generator able to aerosolize the maximum number of viable and culturable cells at elevated particle concentrations. This study performs a comparative investigation of two bioaerosol generators: the Sparging Liquid Aerosol Generator (SLAG) by CH Technologies and the 1520 Flow Focusing Monodisperse Aerosol Generator (FMAG) by TSI. The analysis concentrated on the vitality, culturability, fragmentation, and nebulization efficiency of *E. coli* cells. The results indicated increased fragmentation using the SLAG nebulizer, and the size distribution varied according to the concentration of the injection fluid for FMAG. Both nebulizers imposed significant stress on bacteria during nebulization, halving their viability. Ultimately, the nebulization efficiency of FMAG is twenty times higher than that of SLAG.

## 1. Introduction

Bioaerosols, also referred to as primary biological aerosol particles (PBAPs), constitute airborne biological particles or materials of biological origin suspended in a gaseous medium (Després et al., 2012). PBAPs can impact human health and well-being, though their specific effects remain unclear. The study of these particles has evolved into a multidisciplinary field, combining expertise from physical, biological, and medical sciences. The scientific community maintains a strong interest in bioaerosols due to their profound implications for human health, climate, and various atmospheric processes (Fröhlich-Nowoisky et al., 2016).

An important component of bioaerosols is bacteria (Burrows et al., 2009). Bacteria are found ubiquitously in the atmosphere and play roles in atmospheric processes like cloud formation (Delort et al., 2010), and can also pose risks to human health, including causing respiratory illnesses and allergies (Bolashikov and Melikov, 2009).

Investigating bioaerosols and, in particular, bacteria, in real-world environments is challenging due to the absence of standardized sampling techniques, the biological intricacy and variability of bioaerosols, difficulties in analyzing low concentrations, the complexity of establishing the relationship between exposure and health outcomes, and the swift alterations in bioaerosol properties induced by environmental factors (Blais-Lecours et al., 2015; Mainelis, 2020; Mbareche et al., 2017).

The atmospheric simulation chambers (ASCs) enable the examination of bioaerosols by allowing for precise control and observation of atmospheric conditions, thus facilitating the analysis of bioaerosol behavior and interactions in ways not possible in complex, uncontrolled real-world environments. The ASC enables researchers to manipulate variables,

including temperature, humidity, and gaseous pollutant concentrations, to examine their impacts on the viability and atmospheric behavior of bioaerosols, offering more precise and controlled experimental conditions compared to the unpredictable characteristics of the natural atmosphere (Brotto et al., 2015; Massabò et al., 2018; Vernocchi et al., 2023). Recent years have seen the execution of research involving nebulized microbes in ASCs. A study assessed the disinfection efficiency of Weakly Acidic Hypochlorous Water (WAHW) in an atmospheric simulation chamber against four specific model bacteria (Norkaew et al., 2024). The impact of nitrogen oxides and light on the viability of three distinct bacterial species was examined in a separate study (Gatta et al., 2025).

Previously, these investigations commenced by nebulizing bacteria using aerosol generators whose principal aim was to maximize the quantity of viable and culturable cells aerosolized. The initial process of aerosolization, the nebulization, will impart device-dependent mechanical stress, such as physical shear and wall impaction, resulting from operation of the pressurized system (Stone and Johnson, 2002).

The aerosol generators stress bacteria, and the mechanisms of cellular damage among different nebulizers primarily vary based on their operating principles and how they interact with microorganisms. Bioaerosol generators, in general, exert mechanical stress on bacteria during aerosolization, which can lead to cell membrane rupture and the release of genomic DNA (Danelli et al., 2021; Fahimipour et al., 2018; Liu et al., 2023; Thomas et al., 2011; Zhen et al., 2014a).

The goal of this study was to compare the performance of two aerosol generators in terms of nebulization efficiency, size distribution, bacterial fragmentation, and the stresses produced on bacteria during nebulization, including viability and culturability loss. Although bacterial aerosolization involves multiple sequential stages, including nebulization, aerosol transport, chamber residence, and sampling, these processes are coupled in atmospheric simulation experiments. Therefore, this work adopts a comparative approach, evaluating the two generators under similar and controlled experimental conditions representative of typical chamber-based bioaerosol studies. This methodology enables the identification of generator-specific impacts on bacterial integrity and performance, while preserving experimental relevance for laboratory and simulation chamber applications.

The experiments were conducted within a small-scale aerosol test chamber in stainless steel. This device represents an optimal compromise between a full-scale chamber, as ChAMBRe (Chamber for Aerosol Modelling and Bio-aerosol Research) (Massabò et al., 2018)  and a benchtop setup, specifically designed to investigate bacterial viability within a constrained volume. Functioning as an intermediate simulation chamber, this system significantly facilitates the manipulation of aerosol generators, mechanical equipment, and bacterial suspension. These tests provide a controlled environment for the systematic evaluation of nebulizer performance parameters, specifically including nebulization rate and particle size, thereby facilitating the selection of optimal devices for applications such as developing methods to control biohazardous aerosols, which reached a crescendo during the COVID-19 pandemic, or in general laboratory experiments (Kasler et al., 2025).

## 2. Material and Method

### 2.1 Bacteria strain

For this study, the Gram-negative bacteria *Escherichia coli* (ATCC® 25922™, Thermo Scientific™ Culti-Loops™) were chosen as the bacterial species. These bacteria, with an aerodynamic diameter of around 1 μm (Zhen et al., 2014b), are usually utilized in bioaerosol research, having been proposed as a standard test bacterium (Lee and Kim, 2003; Ruiz and Silhavy, 2022). The *E. coli* suspension for nebulization was prepared by cultivating the bacterial strain in tryptic soy broth (TSB) and incubating it in a shaker incubator (SKI 4 ARGOLAB) at 37 °C. The growth curve was consistently monitored

by measuring the absorbance at $\lambda$ = 600 nm using a spectrophotometer (Shimadzu 1900) until it reached the stationary phase (approximately 1 – overnight bacteria culture (ON)) corresponding to about $10^9$ cells ml$^{-1}$. This approach is a common method for estimating cell culture concentration (Mira et al., 2022). At this stage, 20 ml of the bacterial suspension was centrifuged at 3000 rpm, relative centrifugal force (RCF) 1560 x g, for 10 minutes, using a 12436-centrifuge rotor in an MPW-352 centrifuge (Med Instruments Warsaw, Poland). Finally, the pellet was resuspended in 20 ml of sterile distilled water (MQ). The total bacterial cell concentration was quantified using an automated cell counter (QUANTOM Tx™, Logos Biosystems, QTx). Cell enumeration was performed following staining with Syto 9 fluorescent dye, and the quantification was achieved through automated acquisition and analysis of multiple fluorescence images of the stained cells (Gatta et al., 2025). The viable cells were measured instead by staining the bacteria with LIVE/DEAD BacLight Bacterial Viability Kits (Molecular Probes by Thermo Fisher Scientific™) and capturing fluorescence images using an optical fluorescence microscope (BX43F EVIDENT Europe GmbH). The entire procedure is reported in Gatta et al. 2025. Culturable cells were quantified by enumerating colony-forming units (CFUs) on agar plates. Four Petri dishes were prepared using two distinct dilution levels, with two replicate plates for each dilution. The CFUs were averaged to ascertain the bacterial concentration in the solution, including its statistical uncertainty (standard error of the weighted mean).

## 2.2 Experimental setup

Fig. 1 illustrates the schematic diagram of the experimental setup. The experiments were conducted in a stainless-steel chamber with a volume of approximately 20 litres. Bacterial solution was aerosolized utilizing one of the two aerosol generators evaluated in this study: the Sparging Liquid Aerosol Generator (SLAG, CH Technologies) (Mainelis et al., 2005) and the Flow Focusing Monodisperse Aerosol Generator 1520 (FMAG, TSI) (Duan et al., 2016). The size distribution of the bacterial aerosolization was characterized using the Optical Particle Sizer (OPS) 3330 (TSI) in the optical diameter range of 0.3 to 10 μm, and the WIBS-NEO (Droplet Measurement Technologies®) in the range 0.55 - 10 μm. The aerosolized bacteria were collected with a BioSampler impinger (SKC Inc.) with a 20 ml cup using sterile distilled water as collection fluid. Each sample was collected at the BioSampler nominal flow rate of 12.5 l min$^{-1}$. During the experiment, the nebulization and collection processes were performed simultaneously. At the end of each experiment, the chamber was evacuated by a vacuum pump and subsequently purged with clean air passed through an additional HEPA filter. This cleaning protocol guaranteed that there was no contamination during experiments, as corroborated by the OPS and WIBS readings at the beginning of each experiment (the total and fluorescence particle number at the beginning of all experiments was roughly zero particles cm$^{-3}$). An external connection with a HEPA filter was also installed to maintain the ambient pressure inside the chamber during the bacteria nebulization from the aerosol generator and the collection through the impinger. The BioSampler flow rate was balanced by a HEPA-filtered air inlet to maintain ambient pressure inside the chamber, resulting in short and well-defined aerosol resident time across all experiments.

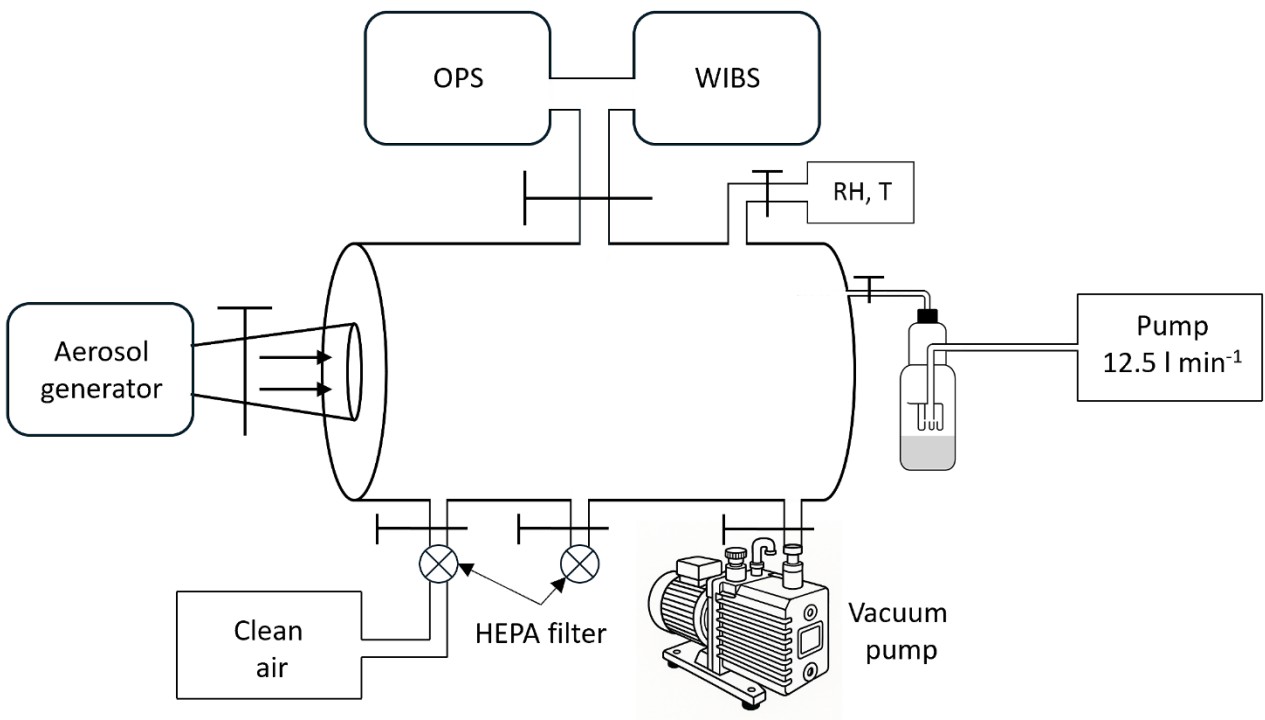

**Figure 1: Experimental setup used to aerosolize *E. coli* with SLAG or FMAG nebulizer (aerosol generator). The particle counter (OPS and WIBS), the cleaning system (clean air and vacuum pump with HEPA filters), and the liquid impinger used to collect the bacteria solution nebulized are also reported.**

### 2.3 Aerosol generator parameters and experimental settings

This study evaluated two aerosol generators: the SLAG and the FMAG. The SLAG (Mainelis et al., 2005) employs a bubbling process that simulates the natural occurrence of bubble bursting to produce particles. The liquid is dispensed with a syringe driver onto a 0.75" diameter porous disk with a nominal pore size of 2 μm. The air is forced from beneath the disk, creating numerous jets. The air jets disintegrate the liquid film into droplets that transport particles, and the smaller droplets eventually exit the device at a designated flow rate. The larger droplets, generated during the process,

gravitationally return to the liquid reservoir and are not aerosolized, nor reintroduced into the circulation.

The FMAG 1520 employs the aerodynamic flow-focusing effect to precisely regulate the diameter of a liquid jet, facilitating the generation of monodisperse droplets ranging from 17 to 90 μm in diameter, which are then dried with a dry dilution airflow (DF) to yield particles measuring between 0.8 and 12 μm in diameter (Duan et al., 2016). During the standard operation, a syringe driver pumps the liquid into the droplet generator; the generated liquid stream exits a 100

125 μm diameter nozzle and is elongated into a significantly thinner filament by the converging gas flow (FF). The resultant thin liquid jet subsequently disintegrates into uniformly sized droplets upon traversing a vibrating (n) ceramic aerosol production head. This modest shear stress typically allegedly allows biological cells to maintain viability, even following dispersion into uniform particles.

For all experiments, the bacterial suspension used for nebulization was prepared by washing the overnight culture and

130 resuspending the cells in MQ (as described in Bacteria strain section), which was used as the injection medium for both SLAG and FMAG. The CFU in the injection solution was approximately $10^9$ CFU mL$^{-1}$, as determined by plate counting prior to nebulization. After aerosolization, CFU concentrations measured in the impinger collection liquid ranged between $10^6$ and $10^7$ CFU mL$^{-1}$, depending on the nebulizer and operating conditions (see Table 2). The aerosol generator parameters selected for our experiments are reported in Table 1: the SLAG settings, used for nebulizing *E. coli*, were

identical to those employed in our prior experiments (Abd El et al., 2024; Agarwal et al., 2024; Gatta et al., 2025; Vernocchi et al., 2023). The duration of the nebulizing process with SLAG and the collecting process with the impinger of each experiment was 20 minutes. For the FMAG, we tested two conditions with the same dilution air flow and flow focusing pressure (to maintain the same pressure inside and outside the chamber during the experiment) but varying the vibration frequencies. The duration of the nebulizing process with FMAG and the collecting process with the impinger of each experiment was 30 minutes. The nebulization durations were selected to ensure that the bacterial count in the impinger collection liquid was above the minimum detection limit of the QTx and the fluorescence microscope ($10^6$ # ml$^{-1}$). For each setting of SLAG and FMAG, three replicated experiments with different *E. coli* strains were performed. To avoid cross-contamination between replicated experiments, at the end of each experiment, the two nebulizers were cleaned with a solution of Ethanol and MQ (70 - 30%) and rinsed with MQ. All experiments were performed at a temperature of (22 ± 1) °C and a relative humidity of (49 ± 1) %, as recorded by a sensor inside the chamber.

**Table 1: SLAG and FMAG parameters setting of each experiment.**

| | SLAG settings | | |
|---|---|---|---|
| | Air flow (l·min$^{-1}$) | Liquid feed rate (ml·min$^{-1}$) | Nebulized solution (ml) |
| EXP_S | 3.5 | 0.4 | 8 |

| | FMAG settings | | | | |
|---|---|---|---|---|---|
| | FF (psi) | DF (l·min$^{-1}$) | ν (Khz) | Liquid feed rate (ml·h$^{-1}$) | Nebulized solution (ml) |
| EXP_F1 | 2.40 | 14.5 | 0 | 4 | 2 |
| EXP_F2 | 2.40 | 14.5 | 200 | 4 | 2 |

## 2.4 Liquid impinger analysis

Upon completion of the experiment, the liquid in the impinger was examined to determine the total, viable, and culturable cell counts. The method employed to assess these three parameters was identical to that utilized for characterizing the nebulized bacterial solution.

## 3. Results and discussion

### 3.1 Methodological framework for data interpretation

The experimental setup, while not permitting a fully decoupled, sequential quantification of each process in bacterial aerosolization, includes numerous design characteristics that facilitate the identification of the primary stressors impacting bacterial survival. The current setup links aerosol generation, chamber resident time, and sampling, mirroring operational settings of atmospheric modeling research. Bacterial culturability and viability were independently assessed in the nebulized solution. Short chamber resident times were ensured, and identical sampling flow was applied for both aerosol generators. In addition, particle size distributions and fragmentation patterns were systematically examined. Under these controlled conditions, the principal differences observed between SLAG and FMAG can be mainly ascribed to the aerosolization phase. This framework provides a basis for interpreting the following results and for identifying generator-specific effects on bacterial integrity. Potential stress associated with aerosol resident time in the chamber and with the

Biosampler sampling is therefore considered a common background contribution, equally affecting all experiments and not biasing the comparative evaluation of the two nebulizers.

### 3.2 Viability and culturability of *E. coli* in sterile distilled water

In these experiments, the *E. coli* bacteria were washed and resuspended in MQ, following a different approach used in our previous research, where the bacteria were resuspended in physiological solution (0.9% NaCl) (Abd El et al., 2024; Agarwal et al., 2024; Gatta et al., 2025; Vernocchi et al., 2023). The physiological solution is ordinarily used for bacterial preparations and suspensions instead of deionized or MQ water because it provides an isotonic environment essential for maintaining bacterial viability and cellular integrity (Omotoyinbo and Omotoyinbo, 2016). However, the resuspension in
MQ eliminates the background signal produced by NaCl particles on the particle counter. It is shown that *E. coli* in sterile distilled/deionized water at room temperature can survive for at least 8 days, even though the viable population count decreases rapidly due to osmotic stress and the lack of nutrients (Aung et al., 2016). The reason is that the bacteria cells exposed to distilled water experience hypotonic stress owing to the absence of solutes in the external environment, resulting in an influx of water that may cause cell enlargement, rupture (osmotic lysis), and subsequent mortality (Wood,
2015) For this reason, we measured the CFU, live, and total *E. coli* bacteria at different time intervals, starting from the resuspension of the bacteria in MQ, to determine how much MQ stresses the *E. coli*. The results are reported in Fig. 2; T=0 represents the *E. coli* data immediately after the bacteria were resuspended in MQ. The results showed that the total and live cell numbers increased over time, reaching approximately $3 \cdot 10^9$ and $2.5 \cdot 10^9$, respectively, within 2 hours. The CFUs number increased in 1 hour and then the number stabilized at a value around $1.5 \cdot 10^9$. The results indicated that *E.*
*coli*, resuspended in MQ, is not stressed during the short time required to run a chamber experiment (less than one hour). The potential loss of bacterial viability and culturability, as measured in the impinger liquid, can be attributed to the nebulization stress induced by nebulizers. Viability and culturability measured in the BioSampler liquid therefore reflect the combined effects of aerosolization, chamber residence, and sampling, which are considered as a common background contribution in comparison of the two nebulizers.

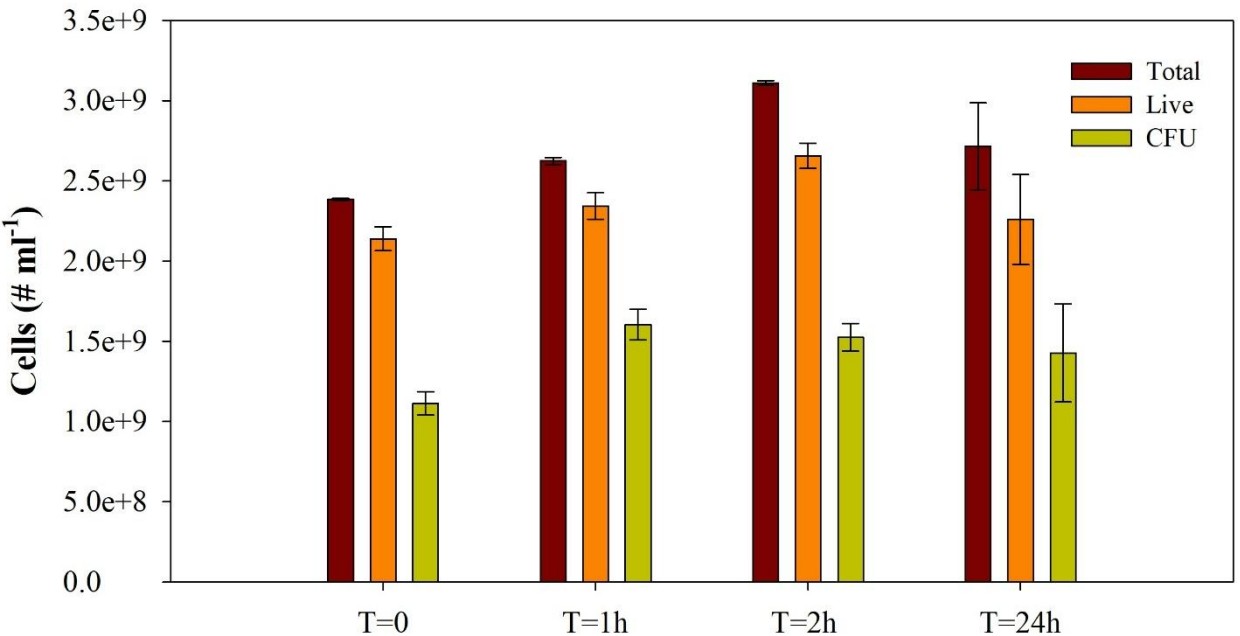

**Figure 2: Total, live, CFU of *E. coli* (# ml⁻¹) after the bacteria resuspension in MQ (T=0) and after 1, 2, and 24 hours.**

### 3.2 Size distribution of nebulized *E. coli* and impinger results

Fig. 3 shows the normalized size distributions of *E. coli* nebulized with FMAG and SLAG. The data in each graph represent the average and standard deviation calculated on the three experiments performed at each nebulizer setting, as detailed in Table 1. The size distribution of FMAG nebulization, assessed using WIBS and OPS, exhibits similarity across the two experimental conditions. Specifically, for EXP_F1, the mean and standard deviation of the Gaussian curve were $(2.8 \pm 0.6)$ μm for WIBS and $(2.6 \pm 0.5)$ μm for OPS. In F2, the values were $(2.9 \pm 0.6)$ μm for WIBS and $(2.8 \pm 0.6)$ μm for OPS. The variable-frequency vibration of the FMAG aerosol generation head does not significantly affect the size distribution of *E. coli* aerosols, although it influences the reproducibility of the results across independent experiments over each setting; in fact, the standard deviation of the data in each size bin of the experiments conducted at zero vibration frequency exceeds that of the experiments at a frequency of 200 kHz. This outcome is expected as the frequency of the vibrating ceramic facilitates the disintegration of the thin liquid jet into minute water droplets (Duan et al., 2016) harboring *E. coli*. The size distribution of the SLAG aerosolization exhibited different characteristics: specifically, the OPS data (fig.4) indicated a peak at the midpoint bin of 0.35 μm and another around 1 μm. The initial peak could be linked to the fragmentation caused by the stress from the aerosolization process, which likely originated from damaged bacteria (Park et al., 2009). When considering particles smaller than 0.5 μm as fragments, this fragmentation accounts for approximately 40% of the total nebulized particles. The WIBS and OPS data, normalized over the same diameter range, exhibited a similar trend, with the WIBS peak slightly shifted, centered around 0.8 μm (fig.5). The peak is higher compared to the OPS peak. When comparing the size distribution of the two nebulizers, it becomes clear that fragmentation is absent in the FMAG experiments, suggesting that FMAG nebulization applies less stress on biological cells. It is worth noting that the size distribution obtained with FMAG and SLAG nebulizers differed: subtracting the fragmentation window, the SLAG experiments exhibited an *E. coli* peak around 0.8 – 0.9 μm, analogous to previous research (Zhen et al., 2014), but FMAG demonstrated a value around three times greater (around 3 μm). The larger particle diameter, achieved with FMAG, may be associated with the concentration of bacteria in the injection solution. When the bacterial concentration is high, during the nebulization process, the bacteria tend to aggregate, resulting in droplets that do not contain a single bacterium. The results corroborated this observation: the WIBS size distribution of *E. coli*, nebulized by FMAG utilizing an ON concentration of 1:100 (ON in MQ, diluted by a factor of 100), using the settings 2, has a peak approximately at 0.8 μm (Fig. 6), similar to the SLAG nebulization. The cell aggregation, occurred during the nebulization, may experience partial shielding from environmental stresses such as dehydration and mechanical damage, potentially enhancing survival compared with isolated single cells (Flemming and Wingender, 2010; Tang, 2009; Vejerano and Marr, 2018).

Table 2 presents the results of total *E. coli* bacteria obtained from the MQ collected by the impinger post-nebulization, detailing the ratio of viable cells and CFU relative to total counts (Live/Tot and CFU/Tot, respectively). Fig. 7 shows the average and standard deviation of nebulization efficiency (NE) throughout duplicated tests with the two nebulizers. The NE is defined as the ratio of the total bacteria collected in the impinger MQ to the total bacteria in the injection solution, divided by the volume of nebulized solution (2 ml for FMAG and 8 ml for SLAG). Because the total and live cells collected with the impinger were less than the minimum detection limit of QTx and the fluorescence microscope, respectively ($10^6$ # cm$^{-3}$), the ON:100 data are not reported. For the FMAG, the two distinct settings yielded comparable Live/Tot and CFU/Tot ratios. Specifically, for the Exp_F1 settings, the average and standard deviation of Live/Tot and CFU/Tot across the three experiments were $(43 \pm 4)$ % and $(19 \pm 4)$ %, respectively. In contrast, for the Exp_F2 settings, the average and standard deviation of Live/Tot and CFU/Tot over the three experiments were $(40 \pm 3)$ % and $(21 \pm 2)$ %. The SLAG exhibited comparable outcomes for the Live/Tot ratio, yet a higher CFU/Tot ratio: the mean and standard

deviation of Live/Tot and CFU/Tot across the three experiments were $(41 \pm 5)$ % and $(33 \pm 7)$ %, respectively. The data obtained from the impinger's collected liquid can be compared with the data from the injection solution. The average and standard deviation of the *E. coli* Live/Tot and CFU/Tot ratios in the injection fluid across the nine studies were $(89 \pm 5)$ % and $(50 \pm 17)$ %, respectively. Both nebulizers diminished the Live/Tot ratio by approximately a factor of 2. In contrast, the CFU/Tot for FMAG in both settings decreased by roughly a factor of 2.5, while the values for SLAG of the injection solution and MQ in the impinger were comparable within the error. Both nebulizers exerted strong stress on the aerosolized *E. coli*, halving the number of live bacteria collected in impingers. Finally, it is important to note how the nebulization efficiency is different between the 2 settings of FMAG and SLAG. In particular, the average and standard deviation of NE were $(0.13 \pm 0.03)$ %, $(0.21 \pm 0.06)$ %, and $(0.008 \pm 0.002)$ %, for FMAG Exp_F1, Exp_F2, and SLAG, respectively. While the NE is similar for both FMAG conditions, the SLAG NE showed a value of about 20 times less than the FMAG NE. The absolute value of NE can be influenced by the geometry of the experimental chamber, in particular, the wall losses (Gatta et al., 2025); even so, it is evident that the FMAG can nebulize bacteria more efficiently than SLAG. This may be attributed to the mechanical stress incurred during the nebulization process and to the distinct nebulization techniques employed by the two nebulizers. As previously demonstrated, the SLAG nebulization produced high fragmentation, reducing the total cells collected in the impinger liquid. In addition, the liquid feed rate of FMAG is very slow (4 ml h$^{-1}$), and all the liquid dispensed in the droplet generator is nebulized. In contrast, the nebulization design of the SLAG system (in which small liquid droplets fall onto a porous septum) generates larger droplets that return to the reservoir rather than being nebulized.

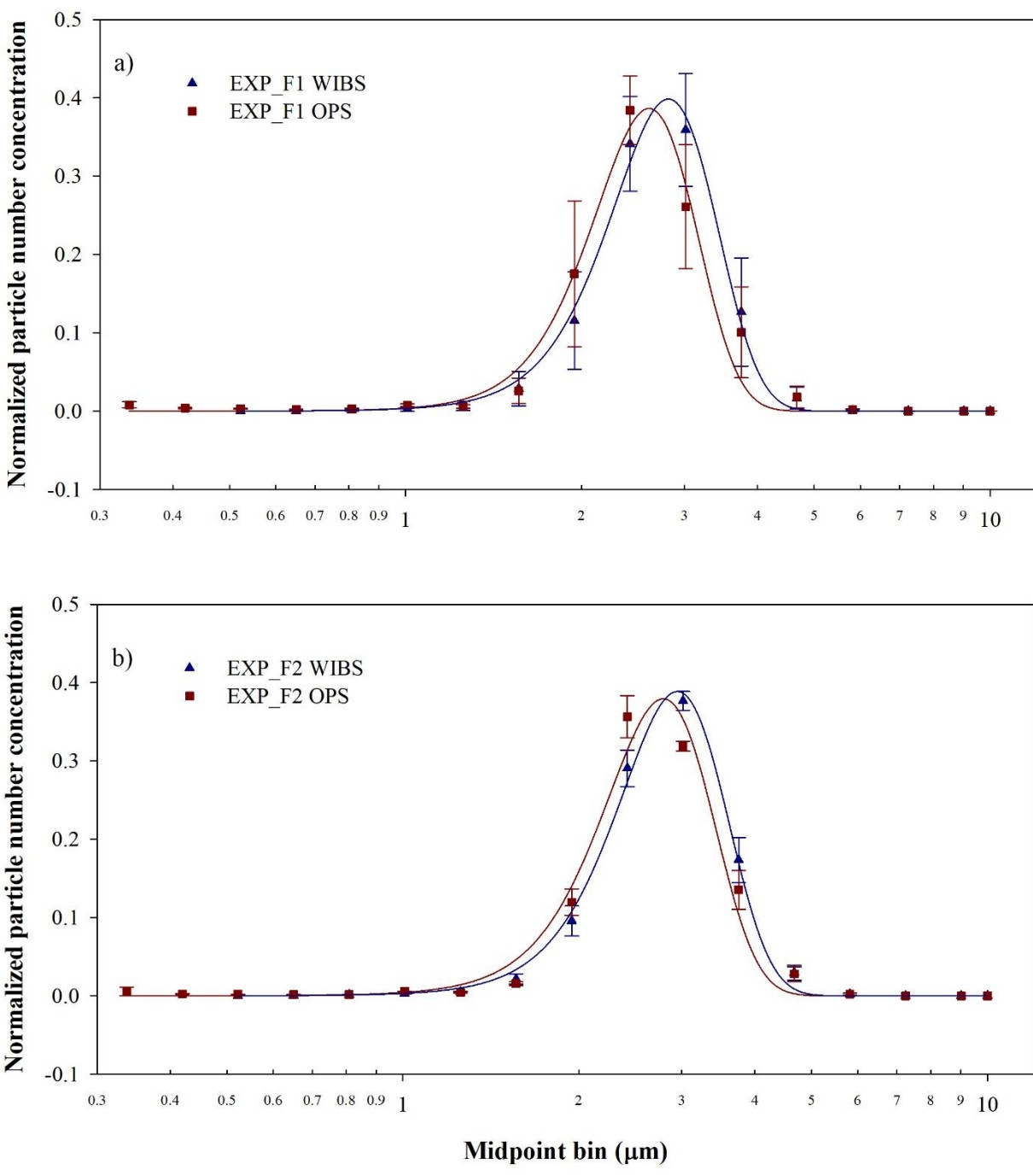

**Figure 3: Normalized size distribution of *E. coli* nebulized using FMAG settings 1 (a) and settings 2 (b) with WIBS and OPS.**

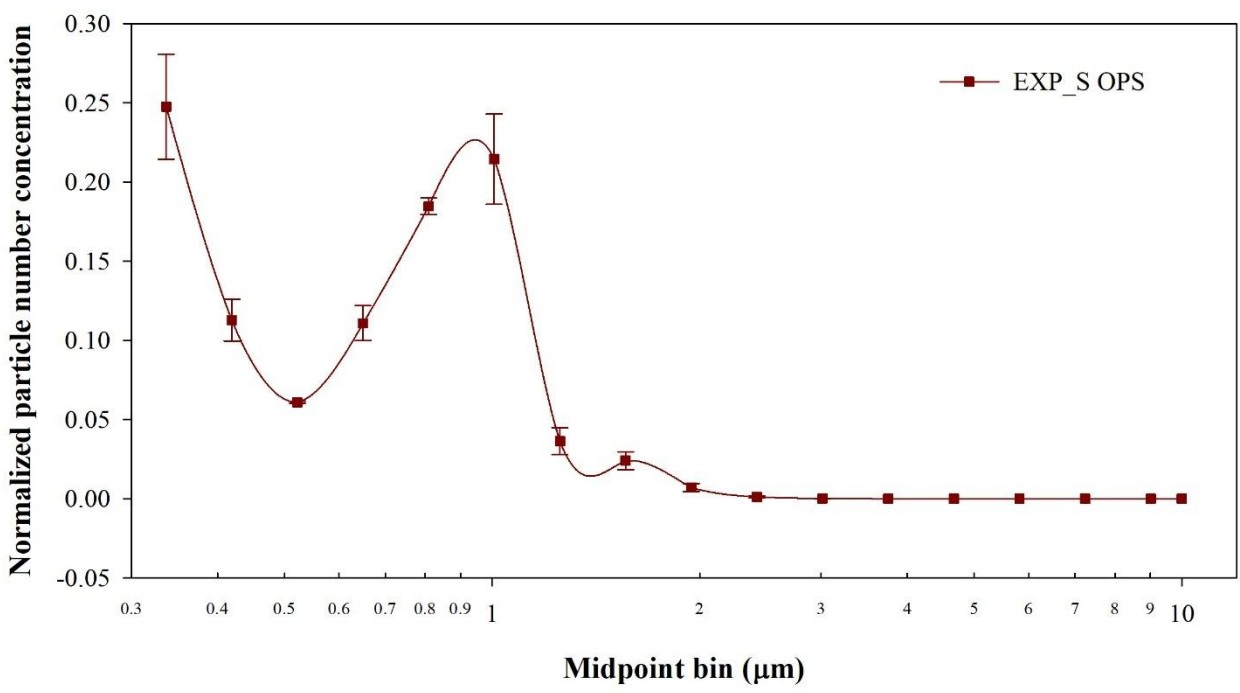

Figure 4: Normalized size distribution of *E. coli* nebulized using SLAG with OPS.

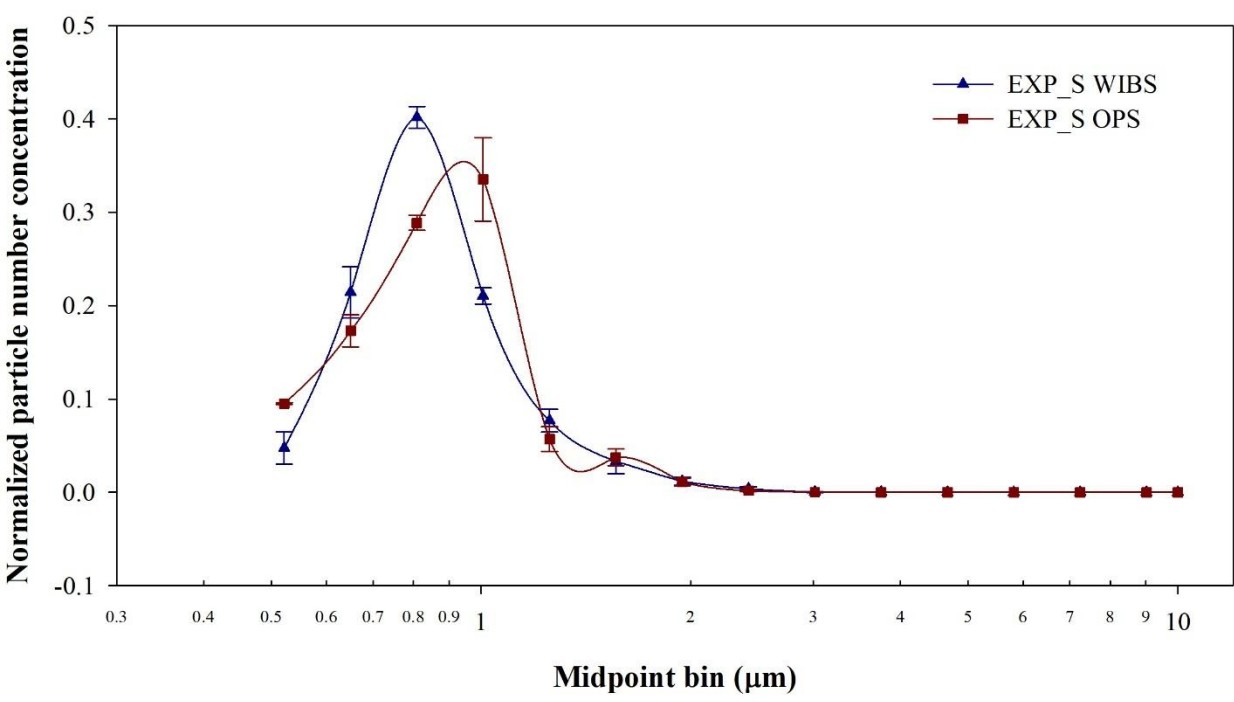

Figure 5: Normalized size distribution of *E. coli* nebulized using SLAG with WIBS and OPS. The data are normalized over the same diameter range, excluding the fragmentation region (OPS region less than 0.5 µm).

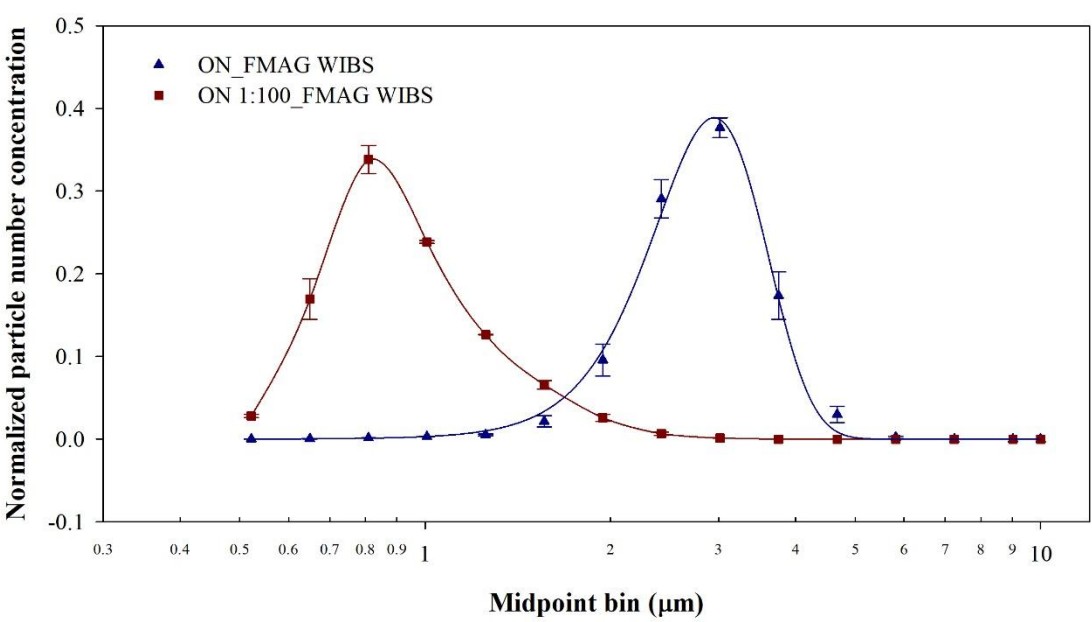

**Figure 6: Normalized size distribution of *E. coli* (ON in MQ in blu and ON 1:100 in MQ in red) aerosolized with FMAG nebulizer using settings F2.**

**Table 2: Total *E. coli* concentration in the injection solution and in the MQ collected with the impinger over different nebulizer settings and experiments. The ratio between live and total, CFU and total bacteria in the liquid impinger is reported. Finally, the average and standard deviation of Live/Tot and CFU/Tot of each nebulizer and settings are reported.**

| | Injection Total cells (# ml$^{-1}$) | Impinger Total cells (# ml$^{-1}$) | Live/Tot Impinger | CFU/Tot Impinger |
|---|---|---|---|---|
| | FMAG Settings F1 | | | |
| EXP1 | $(2.34 \pm 0.04) \cdot 10^9$ | $(4.8 \pm 0.5) \cdot 10^6$ | $(39 \pm 3)$ % | $(16 \pm 2)$ % |
| EXP2 | $(4.6 \pm 0.5) \cdot 10^9$ | $(1.55 \pm 0.01) \cdot 10^7$ | $(44 \pm 2)$ % | $(17 \pm 1)$ % |
| EXP3 | $(4.6 \pm 0.5) \cdot 10^9$ | $(1.18 \pm 0.06) \cdot 10^7$ | $(46 \pm 2)$ % | $(23 \pm 1)$ % |
| **Average ± Std.Dev** | | | **$(43 \pm 4)$ %** | **$(19 \pm 4)$ %** |
| | FMAG Settings F2 | | | |
| EXP1 | $(2.6 \pm 0.2) \cdot 10^9$ | $(8.4 \pm 0.1) \cdot 10^6$ | $(41 \pm 1)$ % | $(19 \pm 2)$ % |
| EXP2 | $(2.45 \pm 0.01) \cdot 10^9$ | $(1.4 \pm 0.5) \cdot 10^7$ | $(37 \pm 2)$ % | $(23 \pm 9)$ % |
| EXP3 | $(2.45 \pm 0.01) \cdot 10^9$ | $(9.1 \pm 0.2) \cdot 10^6$ | $(43 \pm 2)$ % | $(20 \pm 2)$ % |
| **Average ± Std.Dev** | | | **$(40 \pm 3)$ %** | **$(21 \pm 2)$ %** |
| | SLAG settings | | | |
| EXP1 | $(3.0 \pm 0.1) \cdot 10^9$ | $(1.9 \pm 0.1) \cdot 10^6$ | $(43 \pm 12)$ % | $(39 \pm 2)$ % |
| EXP2 | $(3.0 \pm 0.1) \cdot 10^9$ | $(2.4 \pm 0.3) \cdot 10^6$ | $(36 \pm 12)$ % | $(35 \pm 5)$ % |

| | | | | |
|---|---|---|---|---|
| EXP3 | $(4.3 \pm 0.4) \cdot 10^9$ | $(2.5 \pm 0.6) \cdot 10^6$ | $(45 \pm 6)$ % | $(25 \pm 6)$ % |
| **Average ± Std.Dev** | | | **$(41 \pm 5)$ %** | **$(33 \pm 7)$ %** |

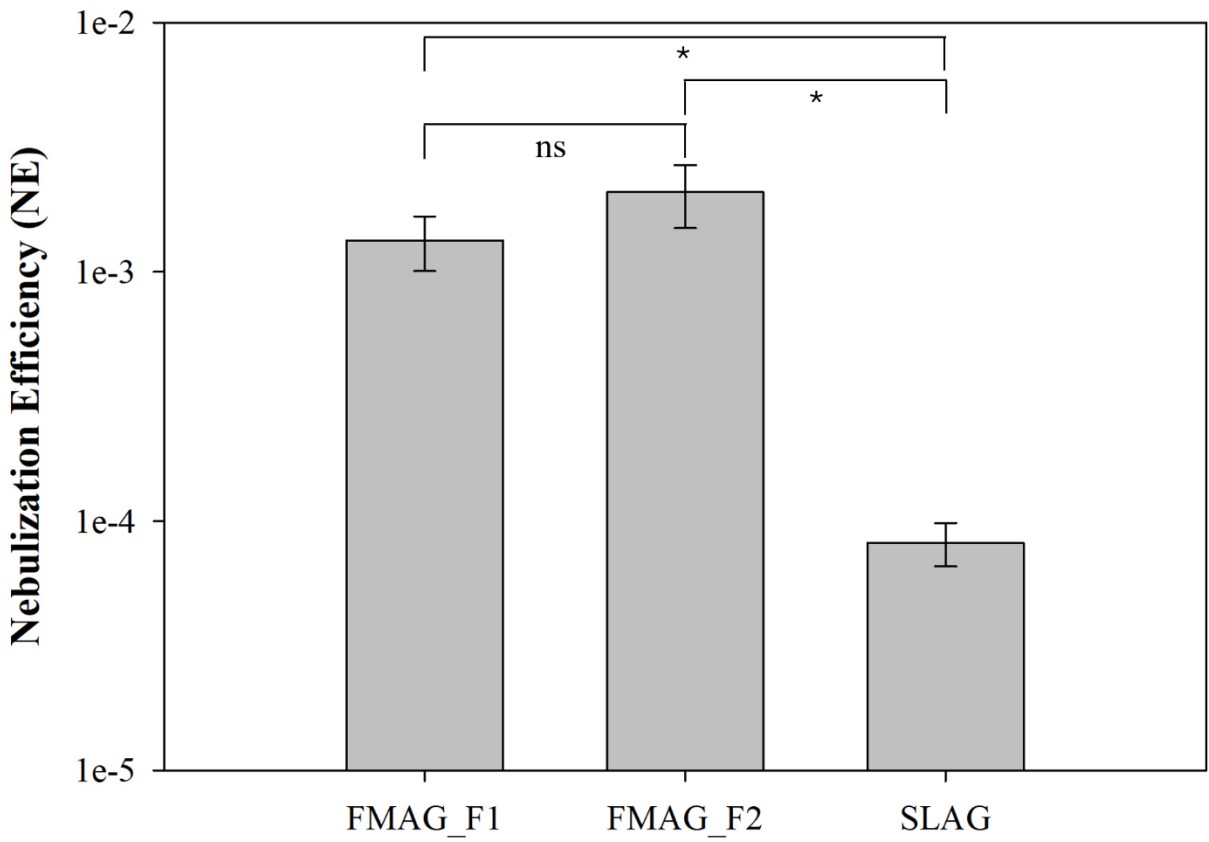

**Figure 7: average and standard deviation of nebulization efficiency of FMAG, in configurations F1 and F2, and SLAG. The P-value of the Student t-test was calculated for each experiment. "ns." stands for "not significant", * p < 0.05.**

## 4. Conclusion

The experimental results indicated that the MQ did not induce stress in the *E. coli* for the duration required to complete the nebulization experiments. The viability and CFU of *E. coli* resuspended in MQ were relatively stable and did not decline within 24 hours following the MQ resuspension of the bacteria. The two FMAG settings yielded identical bacterial size distributions, suggesting that FMAG frequency did not affect size distribution but solely impacted experimental repeatability. The size distribution of FMAG was associated with the bacterial concentration in the injection solution. Utilizing the identical FMAG setup, the nebulization of MQ with ON *E. coli*, diluted by a factor of 100, resulted in a shift of the size distribution peak from roughly 2.8 μm to 0.8 μm. This indicates that with an ON MQ injection liquid (total bacterial concentration > $10^9$ #ml$^{-1}$), the bacteria probably tend to cluster, leading to droplets devoid of individual bacteria. The SLAG nebulizer generated a significant concentration of fragments: using a setting previously employed in our earlier tests (Agarwal et al., 2024; Gatta et al., 2025; Vernocchi et al., 2023), the fragmentation rate is roughly 40% of the total nebulized particles. Both nebulizers induced stress throughout the nebulization process, resulting in a twofold reduction in the viability of the bacteria collected in the impinger. Finally, the FMAG had a nebulization efficiency approximately 20 times greater than that of the SLAG.

This study demonstrated the effect of two different aerosol generators on the aerosolization of bacteria, not only on culturability but also on viability. The compact volume of the chamber represents a choice that prioritizes experimental control and reproducibility over direct representativeness of indoor environments. While larger systems, such as biological safety cabinets or full-scale chambers, allow longer aerosol residence times and closer analogy to ambient indoor conditions, they also introduce additional complexity and variability. The present setup was designed as an intermediate-scale system optimized for comparative evaluation of aerosol generators, rather than for exposure simulation. By isolating generator specific effects under controlled conditions, this work provides a basis for selecting aerosolization techniques in laboratory bioaerosol studies and for interpreting bacterial viability measurements in more complex atmospheric simulations. Future work will extend this approach to Gram-positive bacteria in order to investigate how differences in cell wall structure may influence bacterial response to nebulization in terms of viability, culturability, and fragmentation.

## Data availability

The dataset for this paper can be accessed at DOI: 10.17632/j962hdhc96.1 (Mazzei 2025)

## Author contributions

EG and FM thought up the study and performed the experiments. MB wrote the software to analyze the WIBS data. FM and EG performed all data analysis, with contributions from DM and PP. FP helped with the development of the stainless-steel chamber. EG and FM wrote the manuscript. All authors reviewed and commented on the manuscript.

## Competing interests

The authors declare that they have no competing interests.

## Financial support

This research has been supported by PON PER-ACTRIS-IT (MURIT PON; project no. PIR_00015; "Per ACTRIS IT"), Blue-Lab 60 Net (FESR – Fondo Europeo Di Sviluppo Regionale Azione POR, Regione Liguria, Italy), IR0000032–ITINERIS, Italian Integrated Environmental Research Infrastructures System (D.D. n. 130/2022 - CUP B53C22002150006) funded by the EU (Next Generation EUPNRR, Mission 4 "Education and Research", Component 2 "From research to business", Investment 3.1, "Fund for the realisation of an integrated system of research and innovation infrastructures"), the PNRR MUR Project "Multi Risk sciEnce for resilienT commUnities undeR a changiNg climate (RETURN)" (grant no. PE00000005 CUP HUB B63D22000670006) and the Project 101131261 — IRISCC (Integrated Research Infrastructure Services for Climate Change risks) — HORIZON-INFRA-2023-SERV-01.

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
