# Peer review of "Measurement Report: Effects on viability, culturability, and cells fragmentation of two bioaerosol generators during aerosolization of *E. coli* bacteria"

_EGUsphere, 2025_

## Author Comment (AC1)

This manuscript "Measurement Report: Effects on viability, culturability, and cells fragmentation of two bioaerosol generators during aerosolization of E. coli bacteria" provides valuable insights into bioaerosol sources for simulation chamber–based research. However, several major concerns remain. The authors are encouraged to address the following recommendations to strengthen the clarity, rigor, and interpretability of the manuscript.

Major points:

1) From the title and abstract, I understood that this study aims to provide a comparative investigation of two bioaerosol generators: the Sparging Liquid Aerosol Generator (SLAG) and the Flow Focusing Monodisperse Aerosol Generator (FMAG). However, the manuscript primarily evaluates the performance of the overall system, which includes aerosol generation, aerosol residence within the chamber, and subsequent bioaerosol sampling. As a result, the manuscript lacks a step-by-step evaluation that would help clarify which specific process(es) may be responsible for the observed effects on bacterial viability.

For example, lines 61–68 describe various stresses associated with the use of nebulizers; however, the nebulization step itself should be more explicitly isolated and evaluated.

We thank the reviewer for this thoughtful comment. We agree that the experimental setup evaluates the bioaerosol generation process as a system, including aerosolization, resident time within the chamber, and subsequent sampling, rather than isolating each step independently. Our primary objective was to perform a comparative assessment under similar and realistic experimental conditions, representative of typical chamber-based bioaerosol studies, where aerosol generation, transport, and collection are coupled. For this reason, the performance of SLAG and FMAG was evaluated at the system level rather than through fully decoupled step-by-step tests. Nevertheless, several elements of the experimental design allow us to disentangle the dominant processes responsible for the observed effects on bacterial viability. In particular:

- The stability tests performed on *E. coli* resuspended in MQ water (Fig. 2) demonstrate that neither the suspension medium nor the experimental time scale significantly affects the viability or the culturability of bacteria.
- The aerosol resident time inside the chamber is short for both generators due to the small chamber used and the sampling flow is the same; therefore, the effect of hardware configuration can be considered the same for both nebulizers, and the results can be compared in relative terms.
- The differences observed in particle size distributions and fragmentation rates, especially the presence of a submicron fragmentation mode in SLAG but not in FMAG, provide evidence that mechanical stress during nebulization is an important contributor to bacterial damage.

Based on these observations, we conclude that the comparative approach adopted here allows us to identify aerosol generation as the primary process influencing bacterial viability, with distinct stress mechanisms associated with the two generators.

To better explain the goals of our work, we modified the end of the introduction, and we added a new paragraph to the Results and Discussion section as follows:

Line 68-74: Although bacterial aerosolization involves multiple sequential stages, including nebulization, aerosol transport, chamber residence, and sampling, these processes are coupled in

atmospheric simulation experiments. Therefore, this work adopts a comparative approach, evaluating the two generators under similar and controlled experimental conditions representative of typical chamber-based bioaerosol studies. This methodology enables the identification of generator-specific impacts on bacterial integrity and performance, while preserving experimental relevance for laboratory and simulation chamber applications.

Line 187: "Methodological framework for data interpretation

The experimental setup, while not permitting a fully decoupled, sequential quantification of each process in bacterial aerosolization, includes numerous design characteristics that facilitate the identification of the primary stressors impacting bacterial survival. The current setup links aerosol generation, chamber resident time, and sampling, mirroring operational settings of atmospheric modeling research. Bacterial culturability and viability were independently assessed in the nebulized solution. Short chamber resident times were ensured, and identical sampling flow was applied for both aerosol generators. In addition, particle size distributions and fragmentation patterns were systematically examined. Under these controlled conditions, the principal differences observed between SLAG and FMAG can be mainly ascribed to the aerosolization phase. This framework provides a basis for interpreting the following results and for identifying generator-specific effects on bacterial integrity. Potential stress associated with aerosol resident time in the chamber and with the Biosampler sampling is therefore considered a common background contribution, equally affecting all experiments and not biasing the comparative evaluation of the two nebulizers."

In addition, the remaining liquid within the nebulizer reservoirs after aerosolization should be examined more carefully to assess potential impacts of the nebulization process on bacterial viability.

In the SLAG operating principle, only a fraction of the liquid film disrupted by the air jets is converted into aerosol. Larger droplets generated during the bubbling process revert to the liquid reservoir and are not reintroduced into the aerosol flow and, therefore, don't contribute to the airborne bacterial population. As a result, the residual liquid in the SLAG reservoir is not representative of the aerosolized fraction and cannot be directly used to assess aerosol-phase stress or viability loss. To explain this sentence, we added to the manuscript the following sentence:

 Line 148: The larger droplets, generated during the process, gravitationally return to the liquid reservoir and are not aerosolized, nor reintroduced into the circulation.

2) The physiological status of the bacteria at each experimental step should also be considered more carefully. For instance, lines 290–291 mention … the bacteria probably tend to cluster, leading to droplets devoid of individual bacteria.… Such clustering could lead to prolonged bacterial viability, as cells located within aggregates may be shielded from environmental stresses. This protective effect could also result in higher CFU/mL values. Further clarification and discussion of this possibility would strengthen the interpretation of the results.

We thank the reviewer for this comment. We agree that bacterial clustering may influence physiological status and survival during aerosolization. This aspect has now been explicitly discussed in the revised manuscript

Line 260: The cell aggregation, occurred during the nebulization, may experience partial shielding from environmental stresses such as dehydration and mechanical damage, potentially enhancing survival

compared with isolated single cells (Flemming and Wingender, 2010; Tang, 2009; Vejerano and Marr, 2018).

Flemming, H.-C. and Wingender, J.: The biofilm matrix, Nat Rev Microbiol, 8, 623–633, https://doi.org/10.1038/nrmicro2415, 2010.

Tang, J. W.: The effect of environmental parameters on the survival of airborne infectious agents, J R Soc Interface, 6, https://doi.org/10.1098/rsif.2009.0227.focus, 2009.

Vejerano, E. P. and Marr, L. C.: Physico-chemical characteristics of evaporating respiratory fluid droplets, J R Soc Interface, 15, 20170939, https://doi.org/10.1098/rsif.2017.0939, 2018.

3) The nebulization and sampling periods (20–30 minutes) may themselves contribute to bacterial damage. Prolonged aerosol residence in the chamber, as well as high-velocity airflow into the BioSampler liquid accompanied by vortex motion, may impose additional mechanical stress on bacterial cells. These potential effects should be considered when interpreting the results, as part of an evaluation of the system as a whole.

We thank the referee for this comment. We agree that both aerosol resident time within the chamber and the sampling process itself can contribute to additional mechanical stress on bacterial cells during experiments and should be considered to interpret the results.

To take into account this comment (and comment 2), we added the new paragraph "Methodological framework for data interpretation" as reported above.

Minor points:

Line 92–93: The manuscript states that 20 mL of bacterial suspension was centrifuged at 5000 rpm for 10 minutes. Please specify the applied relative centrifugal force (× g), as rpm alone is insufficient due to rotor-dependent variation. It would also be helpful to clarify whether the potential effects of this centrifugation step on bacterial viability or physiological status were considered.

Thank you for your suggestion. Our intention is to clarify a technical error regarding the rpm reported in the manuscript. The correct value, as specified in our protocol published in Vernocchi et al. (*Atmospheric Measurement Techniques*, 16, 5479–5493, 2023), is 3000 rpm, RCF 1560 x g. This setting was achieved using 12436 centrifuge rotor in the MPW-352 centrifuge model by Med Instruments. The suitability of these parameters is supported by our viability assays under control conditions, as documented in Figure 2 of the manuscript.

We rephrased the line 99 as: At this stage, 20 ml of the bacterial suspension was centrifuged at 3000 rpm, relative centrifugal force (RCF) 1560 x g, for 10 minutes, using a 12436-centrifuge rotor in an MPW-352 centrifuge (Med Instruments Warsaw, Poland). Finally, the pellet was resuspended in 20 ml of sterile distilled water (MQ).

3–114: The particle size range is given as "0.55 ÷ 10 µm." Please confirm whether the symbol "÷" is a typographical error and clarify the intended size range (e.g., 0.55–10 µm).

Thanks for the comment. It was a typo. We intended the size range 0.55 – 10 mm.

Line 147–148: Please specify the medium used to prepare the bacterial suspension for the nebulizers. A brief description of the bacterial solution, including CFU mL$^{-1}$ values before and after the

experiments, would improve transparency and allow better assessment of potential losses during the experimental procedures.

Thanks for the comment. We added the following sentences to the revised manuscript

Line160: For all experiments, the bacterial suspension used for nebulization was prepared by washing the overnight culture and resuspending the cells in MQ (as described in Bacteria strain section), which was used as the injection medium for both SLAG and FMAG. The CFU in the injection solution was approximately $10^9$ CFU mL$^{-1}$, as determined by plate counting prior to nebulization. After aerosolization, CFU concentrations measured in the impinger collection liquid ranged between $10^6$ and $10^7$ CFU mL$^{-1}$, depending on the nebulizer and operating conditions (see Table 2).

Line 149: A nebulization duration of 20 minutes may be sufficiently long to affect bacterial viability. Please clarify whether bacterial viability within the nebulizer reservoir was assessed before and after nebulization. In particular, the SLAG system involves recirculation, which may impose additional stress on bacteria remaining in the reservoir. This potential effect should be considered when interpreting the results.

Thanks for the comment. We note that *E. coli* stability tests in MQ water show no significant loss of viability over time scales longer than the nebulization duration, indicating that liquid-phase residence does not contribute to bacterial damage. In the SLAG system, bacteria remaining in the reservoir are not aerosolized, as previously explained. As mentioned above, we modified line 148 as: "The larger droplets, generated during the process, gravitationally return to the liquid reservoir and are not aerosolized, nor reintroduced into the circulation."

Line 156–157: The manuscript states that all experiments were performed at (22 ± 1) °C and (49 ± 1) % relative humidity. Please clarify whether these values refer to conditions inside the chamber. In addition, indicate how and where temperature and relative humidity were measured, and reflect this information in Figure 1 if appropriate.

Thanks for the comment. Fig.1 was edited to reflect the measurement of T and RH inside the chamber. Finally, the sentence was modified as follows: "All experiments were performed at a temperature of (22 ± 1) °C and a relative humidity of (49 ± 1) %, as recorded by a sensor inside the chamber."

Line 160: The BioSampler uses Milli-Q (MQ) water as the collection liquid and operates with a relatively high airflow rate into the liquid phase, conditions that may impose additional stress on bacterial cells. This potential influence should be discussed when interpreting bacterial viability and culturability results.

Thanks for the comment. We have discussed this aspect in the new paragraph "Methodological framework for data interpretation" as described above.

Line 186-189: ... The results indicated that E. coli, resuspended in MQ, is not stressed during the short time required to run a chamber experiment (less than one hour). The potential loss of bacterial viability and culturability, as measured in the impinger liquid, can be attributed to the nebulization stress induced by nebulizers....

The physiological state of bacteria suspended in MQ water within the BioSampler is likely to differ substantially from the conditions implied by these statements and those represented in Figure 2. It remains unclear whether the bacterial concentrations measured in the BioSampler are directly

comparable to those shown in Figure 2. Clarification on this point is necessary for accurate interpretation of the results. Moreover, the above statements (line 188-189) the suggest that bacterial stress and loss of viability may occur in both the nebulizer and the BioSampler. This overlap in stress sources reduces the ability to clearly attribute observed differences in bacterial viability solely to the nebulization process, thereby weakening the basis for comparing the performance and efficiency of the two nebulizers.

We thank the reviewer for this detailed comment. We agree that the physiological state of *E. coli* suspended in MQ water within the BioSampler differs from that of bacteria maintained in static MQ conditions, as represented in Figure 2. Figure 2 establishes a baseline assessment of bacterial stability in MQ water over time**,** demonstrating that the suspension medium itself does not induce measurable stress on the time scale of the experiments. In addition, since the experimental conditions (i.e., short chamber resident time, temperature, humidity, and sampling flow) are similar, we can also compare the nebulization efficiency of the 2 nebulizers in relative terms.

This aspect is now discussed in the revised manuscript in the introduction section, in the new paragraph of Results and Discussion, and at line 221: "Viability and culturability measured in the BioSampler liquid therefore reflect the combined effects of aerosolization, chamber residence, and sampling, which are considered as a common background contribution in comparison of the two nebulizers."

---

## Author Comment (AC2)

This work investigates the impact of different aerosolization techniques on E.coli within an atmospheric simulation chamber. The topic is surely interesting since it is relevant to many experimental protocols. Also, the analysis is quite thorough and bacteria viability is analyzed in multiple ways to better assess the impact of the aerosolization techniques. The reviewer suggests publication of the paper with only some minor modification as expressed below:

P. 3, Line 85: I suggest reporting also the aerodynamic diameter of E. coli (which should be smaller than 2 um) to make clearer the following choice of 0.5 um as a cut-off for the fragmentation range.

Thanks for the comment. We replaced the typical size with the aerodynamic diameter of *E. Coli* and rephrased as: with an aerodynamic diameter of around 1 μm (Zhen et al., 2014b)

P.5-6: Were the aerosol generators cleaned between replicates to avoid a carry-over effect in bacterial cells?

Thanks for the comment. We added at row 168 the following sentence: To avoid cross-contamination between replicated experiments, at the end of each experiment, the two nebulizers were cleaned with a solution of Ethanol and MQ (70-30%) and rinsed with MQ.

P.5-6: While the droplet diameter of the FMAG is reported (0.8-12 um) the one from the SLAG is not, is there an approximate size of the droplets and is it compatible with E.coli size?

The SLAG is intrinsically designed to generate a polydisperse aerosol through a bubbling and film–jet breakup mechanism. For this reason, a single or nominal droplet diameter is not defined nor reported by the manufacturer, unlike the FMAG, which is specifically designed to produce monodisperse droplets. Consequently, the droplet size generated by the SLAG cannot be uniquely quantified.

P. 9: The WIBS peak is centered around 0.8 um vs. the OPS peak at 0.5 um. Couldn't this be simply an effect on how the different samples define the particles' binning?

We agree with the reviewer that the difference in the peak position is primarily related to the different lower size detection limits and binning schemes of the two instruments. In particular, the OPS measures particles starting from 0.3 μm, while the WIBS detection range begins at 0.55 μm. As a consequence, particles contributing to the smaller-size bins are only detected by the OPS, which shifts the apparent modal diameter toward smaller sizes compared to WIBS.

P. 9: Beside peak shifting WIBS seems to exhibit a higher (i.e.: non-overlapping error bars) maximum normalized concentration compared with OPS. Is this related to total particles counted by WIBS or to fluorescent-only particles? If so is that explainable by the dimensional shift or what alse could explain a higher number of fluorescent particles vs. total ones?

Thanks for the comment. Since only bacteria suspended in MQ are nebulized, all particles observed by WIBS are of biological origin; therefore, total and fluorescent particles counts measured by WIBS coincide. This difference can be attributed to the different size ranges of the two instruments, as explained in the previous comment. This difference in detectable size range and binning can therefore influence the shape and height of the normalized size distributions.

P. 10. Figure 3 is missing x-axis description (Midpoint bin (um)).

Thanks for the comment. We modified the figure following the suggestion, thanks

10P. 14, L. 297-301. While the presented study is surely valuable and a thorough characterization, it is far from being the first step in this field. For example Thomas et al. (2011) already provided data on survival and site of damage of E. coli nebulized with different techniques. Rather, I think that the major

strength of this work is the completeness of the analysis which goes way beyond simple viability (in a culturable sense). I suggest a rephrasing of this sentence.

Thanks for the comment. We added the reference to the manuscript and, considering also the comments of the other reviewers, we modified the conclusions as:

Line 337: This study demonstrated the effect of two different aerosol generators on the aerosolization of bacteria, not only on culturability but also on viability. The compact volume of the chamber represents a choice that prioritizes experimental control and reproducibility over direct representativeness of indoor environments. While larger systems, such as biological safety cabinets or full-scale chambers, allow longer aerosol residence times and closer analogy to ambient indoor conditions, they also introduce additional complexity and variability. The present setup was designed as an intermediate-scale system optimized for comparative evaluation of aerosol generators, rather than for exposure simulation. By isolating generator specific effects under controlled conditions, this work provides a basis for selecting aerosolization techniques in laboratory bioaerosol studies and for interpreting bacterial viability measurements in more complex atmospheric simulations. Future work will extend this approach to Gram-positive bacteria, in order to investigate how differences in cell wall structure may influence bacterial response to nebulization in terms of viability, culturability, and fragmentation.

Finally, this is not a comment on the quality of the paper itself, just an potential outlook for future work. Gram negative and gram positive bacteria differ in their structure, it would be interesting in the future to see if this also translates in differences when nebulized (in terms of viability, etc.).

We thank the reviewer for this valuable and forward-looking suggestion. We fully agree that structural differences between Gram-negative and Gram-positive bacteria, particularly in cell wall composition and thickness, may lead to different responses to nebulization-induced stress, potentially affecting viability, culturability, and fragmentation.

While the present study focuses on *Escherichia coli* as a well-established model organism, extending the analysis to Gram-positive bacteria represents a natural and highly relevant direction for future work, which we plan to address in subsequent studies.

We have added the following sentence to the Conclusion section:

Future work will extend this approach to Gram-positive bacteria, in order to investigate how differences in cell wall structure may influence bacterial response to nebulization in terms of viability, culturability, and fragmentation

Cited literature:

Thomas RJ Webber D, Hopkins R, Frost A, Laws T, Jayasekera PN, Atkins T.2011.The Cell Membrane as a Major Site of Damage during Aerosolization of Escherichia coli . Appl Environ Microbiol. 77:.https://doi.org/10.1128/AEM.01116-10

---

## Author Comment (AC3)

The introduction sets the context of the study effectively, in particular the lack of standardised approaches for bioaerosol research and the impact of aerosolisation and bioaerosol sampling methods on the viability of airborne microorganisms. As with other areas of environmental microbiology, the issues of mechanical sampling stresses, also the question of viable, but non-culturable microorganisms and the biases these effects can bring to data collection, remain a challenge for aerobiologists. Any information concerning bioaerosol generation and its influence on experimental findings is therefore helpful.

Lines 84-87 – General comment - As a test organism, the use of an *E. coli* strain was understandable, if challenging, given the relatively lower robustness of Gram-negative, nonspore-forming bacteria compared to the Gram-positive species often used in aerosol test studies.

Thanks for the comment. Our *E. coli* bacterial strain (ATCC 25922; American Type Culture Collection, Manassas, VA, USA) was selected as the model organism for this study. E. coli is a premier bacterial model due to its rapid growth rate and cost-effective cultivation requirements. Furthermore, it can be handled under Biosafety Level 1 (BSL-1), conditions using standard laboratory practices. Our future goal is to adapt this protocol for other significant bioaerosol studies involving different microbial species.

Moreover, the endotoxins, components of the outer membrane of gram-negative bacteria are significant bioaerosol constituents known to trigger respiratory issues in exposed individuals. So could be interesting to investigate this cellular component of the outer membrane of the cell wall consisting of lipids and lipopolysaccharides (LPS). Endotoxins are found in high concentrations in the air at sites that handle organic material such as composting facilities, intensive farms, and wastewater operations.

- Wouters, I.M.M. et al. "Overview of Personal Occupational Exposure Levels to Inhalable Dust, Endotoxin, β (1→3)-Glucan and Fungal Extracellular Polysaccharides in the Waste Management Chain". Ann. Occup. Hyg. 2005, 50, 39–53.
- Lawniczek-Walczyk, A. et al. "Occupational exposure to airborne microorganisms, endotoxins and beta-glucans in poultry houses at different stages of the production cycle". Ann. Agric. Environ. Med. 2013, 20, 259–268.
- Thorn, J et al. "Measurement Strategies for the Determination of Airborne Bacterial Endotoxin in Sewage Treatment Plants". Ann. Occup. Hyg. 2002, 46, 549–554.

Line 93 – as mentioned by at least one other reviewer, it would be helpful to have the centrifugation conditions expressed as xg to present a standardized value and to allow reproducibility, should the method be applied by others across different machines.

Thank you for your suggestion. Our intention is to clarify a technical error regarding the rpm reported in the manuscript. The correct value, as specified in our protocol published in Vernocchi et al. (*Atmospheric Measurement Techniques*, 16, 5479–5493, 2023), is 3000 rpm, RCF 1560 x g. This setting was achieved using 12436 centrifuge rotor in the MPW-352 centrifuge model by Med Instruments. The suitability of these parameters is supported by our viability assays under control conditions, as documented in Figure 2 of the manuscript.

We rephrased the line 99 as: At this stage, 20 ml of the bacterial suspension was centrifuged at 3000 rpm, relative centrifugal force (RCF) 1560 x g, for 10 minutes, using a 12436-centrifuge rotor in an MPW-352 centrifuge (Med Instruments Warsaw, Poland). Finally, the pellet was resuspended in 20 ml of sterile distilled water (MQ).

Lines 107-108 – the test chamber was small at 20l. Even a class II biological safety cabinet would have been more specious and perhaps more representative of an ambient indoor atmosphere where particles can remain airborne at least for minutes prior to collection. BSCs can also be cleaned very effectively, have HEPA filtered inlet air and have a reasonable internal volume, typically, of 0.8 to 1.0m$^3$. If not described elsewhere it would be useful if any perceived limitations of this very compact chamber choice were presented, compared with other obvious options such as a BSC, or just a larger steel vessel of the type used. Maybe this could be added to the Conclusions section - see comment below?

We thank the reviewer for this comment. We agree that the relatively small volume of the test chamber (≈20 L) represents a methodological choice that differs from larger systems such as Class II biological safety cabinets or large-scale atmospheric simulation chambers.

The compact chamber was selected as a compromise between benchtop setups and large-volume facilities, with the primary objective of enabling a highly controlled and reproducible comparison of aerosol generators under well-defined conditions. The reduced volume minimizes wall-cleaning and decontamination times and facilitates systematic replication while maintaining full control over resident time, humidity, and sampling conditions.

To respond to this comment, we added the following sentence to the conclusion:

"The compact volume of the chamber represents a methodological choice that prioritizes experimental control and reproducibility over direct representativeness of indoor environments. While larger systems, such as biological safety cabinets or full-scale chambers, allow longer aerosol residence times and closer analogy to ambient indoor conditions, they also introduce additional complexity and variability. The present setup was designed as an intermediate-scale system optimized for comparative evaluation of aerosol generators, rather than for exposure simulation.

Line 113 – I presume that the aerosol range should read, "...in the range 0.55 **-**10μm", rather than "....in the range 0.55 ÷ 10μm."

Thanks for the comment. It was a typo. We intended the size range 0.55 – 10 μm.

Line 115 and elsewhere – It would be useful to have the rationale for the choice of sterile water (for E. coli suspension and bioaerosol collection). I can see that this might eliminate any risk of crystal formation, which might interfere with some of the applied assays, but the use of freshly prepared, sterile isotonic buffer (such as phosphate buffered saline) would perhaps have conferred increased protection for the aerosolised and sampled particles, and is perhaps the cell suspension medium that many others would have preferred. Having read on, I do note that there is further comment on this between lined 169 and 189 and the journal editor may feel that this explanation is sufficient. Figure 2 does go some way to indicating that viability was retained within the same order of magnitude (recoverable CFU/ml), throughout the short course of the experimentation.

Thanks for the comment. We leave it to the Editor to judge if our explanation is enough.

Line 116 – The sampling pump flow rate is given as 12.5 l/min. Given the limited test chamber volume did this have any implications for the testing? For sample, very limited exposure of the bioaerosols to the airborne state prior to sampler entrapment? From the test setup diagram I assume that any negative pressure effects in the chamber were avoided by use of the balancing effect of the HEPA filtered air inlet, but please comment on this further on the set-up if you can.

We thank the reviewer for this comment. We acknowledge that, given the limited chamber volume, the BioSampler flow rate implies relatively short aerosol resident times before collection. This experimental condition was selected to enable rapid attainment of steady-state concentrations and to ensure sufficient bacterial recovery above detection limits. And yes, the chamber was operated under pressure-balanced conditions. This configuration ensured stable aerosol conditions and avoided additional stress related to pressure gradients.

In the present study, aerosol resident time and sampling conditions are treated as systematic background contributions, common to all measurements, and we have clarified these aspects in the revised manuscript to better describe the implications of chamber volume, sampling flow rate on the experimental conditions (the new paragraph in Results and Discussion called "Methodological framework for data interpretation").

Finally, for a better understanding, we added the following sentence at Line 133: The BioSampler flow rate was balanced by a HEPA-filtered air inlet to maintain ambient pressure inside the chamber, resulting in short and well-defined aerosol resident time across all experiments.

Lines 144-145 – Is this assumption based on the manufacturer's information, or a fact established by other independent evaluation? It would be useful to have that qualified in the text, or to modify the sentence in a way such as, "This modest shear stress typically **allegedly** allows biological cells to maintain viability, even following dispersion into uniform particles."

We agree with the referee. We modified the text as suggested

Figure 3 required additional labelling on the x-axis - at least one other reviewer has commented on this.

Thanks for the comment. We modified figure 3 by adding the labelling on the x-axis

Lines 297-301 - Conclusions section – are there any lessons learnt in terms of any experimental weaknesses identified? The MQ water issue is well outlined, but there are few comments on experimental weaknesses or biases otherwise. Also, the starting volume for each nebuliser is quite different – do the authors have any comments on the implications of this for wider aerosolisation testing and the practical choices to be made? The paper of Gatta et al., 2025 is referred to several times throughout this manuscript; but are there any comparisons or reflections to be drawn between this earlier paper and the new manuscript. As this earlier published work is clearly important to the current paper, it would be useful to get that perspective

Thanks for the comment. We have revised the conclusion, and here is the new version:

The experimental results indicated that the MQ did not induce stress in the *E. coli* for the duration required to complete the nebulization experiments. The viability and CFU of *E. coli* resuspended in MQ were relatively stable and did not decline within 24 hours following the MQ resuspension of the bacteria. The two FMAG settings yielded identical bacterial size distributions, suggesting that FMAG frequency did not affect size distribution but solely impacted experimental repeatability. The size distribution of FMAG was associated with the bacterial concentration in the injection solution. Utilizing the identical FMAG setup, the nebulization of MQ with ON *E. coli*, diluted by a factor of 100, resulted in a shift of the size distribution peak from roughly 2.8 μm to 0.8 μm. This indicates that with an ON MQ injection liquid (total bacterial concentration > $10^9$ #ml$^{-1}$), the bacteria probably tend to cluster, leading to droplets devoid of individual bacteria. The SLAG nebulizer generated a significant concentration of fragments: using a setting previously employed in our earlier tests (Agarwal et al., 2024; Gatta et al., 2025;

Vernocchi et al., 2023), the fragmentation rate is roughly 40% of the total nebulized particles. Both nebulizers induced stress throughout the nebulization process, resulting in a twofold reduction in the viability of the bacteria collected in the impinger. Finally, the FMAG had a nebulization efficiency approximately 20 times greater than that of the SLAG.

This study demonstrated the effect of two different aerosol generators on the aerosolization of bacteria, not only on culturability but also on viability. The compact volume of the chamber represents a choice that prioritizes experimental control and reproducibility over direct representativeness of indoor environments. While larger systems, such as biological safety cabinets or full-scale chambers, allow longer aerosol residence times and closer analogy to ambient indoor conditions, they also introduce additional complexity and variability. The present setup was designed as an intermediate-scale system optimized for comparative evaluation of aerosol generators, rather than for exposure simulation. By isolating generator specific effects under controlled conditions, this work provides a basis for selecting aerosolization techniques in laboratory bioaerosol studies and for interpreting bacterial viability measurements in more complex atmospheric simulations. Future work will extend this approach to Gram-positive bacteria, in order to investigate how differences in cell wall structure may influence bacterial response to nebulization in terms of viability, culturability, and fragmentation.